# A Simple Kinematic Model for the Lagrangian Description of Relevant Nonlinear Processes in the Stratospheric Polar Vortex

Victor José García-Garrido[1], Jezabel Curbelo[1,2], Carlos Roberto Mechoso[3], Ana María Mancho[1], and Stephen Wiggins[4]

[1]Instituto de Ciencias Matemáticas, CSIC-UAM-UC3M-UCM. C/ Nicolás Cabrera 15, Campus de Cantoblanco UAM, 28049 Madrid, Spain.
[2]Departamento de Matemáticas, Facultad de Ciencias, Universidad Autonóma de Madrid, 28049 Madrid, Spain.
[3]Department of Atmospheric and Oceanic Sciences, University of California at Los Angeles, Los Angeles, California.
[4]School of Mathematics, University of Bristol. Bristol BS8 1TW, UK.

*Correspondence to:* A. M. Mancho (a.m.mancho@icmat.es)

**Abstract.** In this work we study the Lagrangian footprint of the planetary waves present in the Southern Hemisphere stratosphere during the exceptional Sudden Stratospheric Warming event that took place during September 2002. Our focus is on constructing a simple kinematic model that retains the fundamental mechanisms responsible for complex fluid parcel evolution, during the polar vortex breakdown and its previous stages. The construction of the kinematic model is guided by the Fourier decomposition of the geopotential field. The study of Lagrangian transport phenomena in the ERA-Interim reanalysis data highlights hyperbolic trajectories and these trajectories are Lagrangian objects that are the kinematic mechanism for the observed filamentation phenomena. Our analysis shows that the breaking and splitting of the polar vortex is justified in our model by the sudden growth of a planetary wave and the decay of the axisymmetric flow.

## 1 Introduction

The availability of high-resolution and high-quality reanalysis data sets provides us with a powerful tool for obtaining a detailed view of the space-time evolution of the stratospheric polar night vortex (SPV), which has implications for the geophysical fluid dynamics of the entire earth. The complexity of such a detailed view, however, makes it difficult to extract the physical mechanisms underlying notable transport features in the observed behaviour. The goal of this work is to gain new insights into the fundamental mechanisms responsible for complex fluid parcel evolution, since these lie at the heart of our understanding of the dynamics and chemistry of the stratosphere. To this end we extract, directly from the data, a simple model with a stripped-down dynamics in order to directly probe, in a controlled and systematic manner, the physical mechanisms responsible for the key observed transport features of the SPV. Models of this kind, termed "kinematic models" have provided a simple approach for studying Lagrangian transport and exchange associated with flow structures such as meandering jets and travelling waves (Bower, 1991; Samelson, 1992; Malhotra and Wiggins, 1998; Samelson and Wiggins, 2006). Other works have used analytical kinematic models to illustrate phenomena in planetary atmospheres (e.g., Rypina *et al.* (2007); Morales-Juberías *et al.* (2015)).

In the present paper, we focus on SPV transport processes associated with filamentation and vortex breaking, of which the dynamical structure is not fully understood.

The importance of an increased understanding of the SPV was dramatically demonstrated by the intense research effort that followed the discovery of the "Antarctic Ozone Hole" phenomenon in the 1970's (Chubachi, 1984; J.D.Farman *et al.*, 1985; Solomon, 1988). Following decades during which monitoring of ozone in atmospheric columns above Antarctica showed little interannual variability, in situ measurements corroborated by satellite data, revealed that ozone was systematically decreasing in the Antarctic lower stratosphere during the southern spring season. Whilst this was immediately associated with the simultaneous increase in atmospheric pollution by anthropogenic activities, several key questions arose (Solomon, 1999): 1) Why over Antarctica and not over the Arctic since pollution sources are stronger in the northern than in the southern hemisphere? 2) Why in the spring season? and 3) Will ozone depletion extend worldwide? The research demonstrated that, indeed, increased atmospheric pollution was to be blamed for the ozone depletion and identified the participating substances and special mechanisms. The research also demonstrated that the unique atmospheric conditions above Antarctica were responsible for the geographic preference for ozone destruction. In particular, it was shown that the strong circumpolar and westerly SPV characteristic of the southern winter and spring stratosphere contributes to isolate the cold polar region, setting up a favorable environment for the special chemistry to act. The new knowledge led to the formulation of international agreements that resulted in a negative answer to question 3) above. The analysis of transport of fluid parcels outside the region isolated by the SPV, also showed strong stirring and mixing of the flow. In this "surf zone" (McIntyre and Palmer, 1984) air parcels can travel long distances away from the SPV in an environment where contours of long-lived tracers, such as potential vorticity, can stretch forming complex patterns. In this region, Rossby wave breaking is associated with irreversible deformation that pulls material filaments of the outer edge of the SPV and enhances mixing with the exterior flow (McIntyre and Palmer, 1983, 1984, 1985). Such a process makes the SPV edge a barrier to horizontal transport of air parcels (Juckes and McIntyre, 1987) while continuously eroding and regenerating the SPV edge by filamentation (Bowman, 1993). Polvani and Plumb (1992) and Nakamura and Plumb (1994) examined in an idealized setting the way in which Rossby waves break ejecting SPV material outward. The latter conceived a similar setting in which Rossby waves break also inwards.

Dynamical systems theory provides valuable insights into the transport processes described in the previous paragraph. Tools of the theory include the geometrical structures referred to as hyperbolic trajectories (HTs), their stable and unstable manifolds and their intersection in homoclinic and heteroclinic trajectories that provide the theoretical and computational basis for describing the filamentation process. A challenge in the application of these concepts to realistic geophysical flows is that while the structures mentioned are defined for infinite time autonomous or periodic systems, geophysical flows are typically defined as finite-time data sets and are not periodic. Mancho *et al.* (2006b) addressed this challenge for realistic ocean flows by identifying special hyperbolic trajectories in the finite data set, called distinguished hyperbolic trajectories (DHT), and by computing stable and unstable manifolds as curves advected by the velocity field. A pioneering effort for identifying HT for the stratosphere was due to Bowman (1993). McIntyre and Palmer (1983); Bowman (1996); de la Cámara *et al.* (2013) suggested that HTs are responsible for the cat's eye structures associated with planetary wave breaking at the critical levels, i. e. where the wave phase speed matches the background velocity (Stewartson, 1977; Warn and Warn, 1978). HTs are at the locations

where the cats eyelids meet. Perturbation of the cat's-eyes results in irreversible deformation of material contours, signifying Rossby wave breaking. de la Cámara *et al.* (2013) and Guha *et al.* (2016) identified HTs both within and outside the SPV, thus suggesting that Rossby wave breaking can occur in either of those regions. The former authors worked with reanalysis data, while Guha *et al.* (2016) used a dynamical model based on the shallow water equations in which the perturbing waves are produced in a controlled manner. Therefore, HTs are essential features for tracer mixing both outside and inside the vortex, and for occasional air crossings of the vortex edge.

We focus on the SPV behavior during the major stratospheric sudden warming that occurred in the southern stratosphere during September 2002. In this unusual event, the SPV broke down in the middle stratosphere (Mechoso *et al.*, 1988; Varotsos, 2002, 2003, 2004; Allen *et al.*, 2003; Konopka *et al.*, 2005; Esler and Scott, 2005; Manney *et al.*, 2006; Charlton *et al.*, 2006; Taguchi, 2014). We begin by identifying key Lagrangian features of the flow in reanalysis data fields. Next we build a kinematic model of the event, that emulates the behavior of planetary waves observed in the data. We show that our model produces strikingly similar transport features to those found in the reanalysis data, confirming the key role played by the HTs during vortex filamentation and breakdown.

The structure of the paper is as follows. Section 2 describes the data and methods we use. Section 3 describes the planetary waves in the reanalysis data in the year 2002 in the stratosphere at selected pressure levels (10hPa). and we relate these to filamentation phenomena and the polar vortex breakdown that occurred in that year. Section 4 reproduces the findings obtained with our analytical kinematical model confirming the role played by the HTs in the 2002 vortex filamentation and breakdown. Section 5 discusses the consistency of the kinematic model as representative of atmospheric flows that conserve potential vorticity. Finally, in section 6 we present the conclusions.

## 2    Data and Methods

### 2.1    ERA-Interim Reanalysis Data

To achieve a realistic representation of the atmospheric transport processes, it is crucial to use a reliable and high-quality dataset. We use in this work the ERA-Interim reanalysis dataset produced by a weather forecast assimilation system developed by the European Centre for Medium-Range Weather Forecasts (ECMWF; Simmons *et al.* (2007)). de la Cámara *et al.* (2013) obtained encouraging results on the suitability of the ERA-Interim dataset for Lagrangian studies of stratospheric motions in their comparison of parcel trajectories on the 475 K isentropic surface (around 20 km) using this dataset and the trajectories of superpressure balloons released from Antarctica by the VORCORE project during the spring of 2005 (Rabier *et al.*, 2010).

The Era-Interim data set that we selected for this study is available four times daily (00:00 06:00 12:00 18:00 UTC), with a horizontal resolution of $1° \times 1°$ in longitude and latitude and 60 sigma levels in the vertical from 1000 to 0.1 hPa. The data covers the period from 1979 to the present day (Dee *et al.*, 2011) and it can be downloaded from http://apps.ecmwf.int/datasets/data/interim-full-daily/levtype=sfc/. In particular we will use the data for the geopotential height and wind fields on surfaces of constant pressure for the period August-September 2002.

The geopotential height $Z$ on constant pressure surfaces $p$ is defined as the normalization to $g_0 = 9.80665 \, ms^{-2}$ (standard gravity at mean sea level) of the gravitational potential energy per unit mass at an elevation $s$ (over the Earth's surface), and has the form:

$$Z(\lambda, \phi, p, t) = \frac{1}{g_0} \int_0^{s(p,t)} g(\lambda, \phi, z) \, dz \, , \tag{1}$$

where $g$ is the acceleration due to gravity, $\lambda$ is longitude, $\phi$ is latitude and $z$ is the geometric height (Holton (2004)). In the quasi-geostrophic approximation, the geopotential height is proportional to the streamfunction of the geostrophic flow (Holton, 2004).

For the analysis of planetary waves, we apply a zonal Fourier decomposition to the geopotential height field on the 10 hPa pressure level (approximately 850 K potential temperature). The zonal wave decomposition yields:

$$Z = \mathcal{Z}_0(\phi, p, t) + \sum_{k=1}^{\infty} \mathcal{Z}_k(\lambda, \phi, p, t) \, . \tag{2}$$

The mean flow is defined as:

$$\mathcal{Z}_0(\phi, p, t) = \frac{1}{2\pi} \int_0^{2\pi} Z(\lambda, \phi, p, t) \, d\lambda \, , \tag{3}$$

and the different modes $\mathcal{Z}_k$ with wavenumber $k \geq 1$ have the sinusoidal description:

$$\mathcal{Z}_k(\lambda, \phi, p, t) = \mathcal{B}_k(\phi, t) \cos(k\lambda + \varphi_k(\phi, p, t)) \tag{4}$$

where $\lambda \in [0, 2\pi)$ is longitude, $\phi \in [-\pi/2, \pi/2]$ is latitude, $\mathcal{B}_k$ is the amplitude of the wave and $\varphi_k$ its phase. During the warming event occurred in the southern stratosphere during September 2002 the flow was dominated by the contributions of the mean flow and the two longest planetary waves ($\mathcal{Z}_1$ and $\mathcal{Z}_2$; Krüger *et al.* (2005))

## 2.2 Lagrangian Descriptors

Dynamical systems theory provides a qualitative description of the evolution of particle trajectories by means of geometrical objects that partition the phase space (the atmosphere in our case) into regions in which the system shows distinct dynamical behaviors. These geometrical structures act as material barriers to fluid parcels and are closely related to flow regions known as hyperbolic, where rapid contraction and expansion takes place. Several Lagrangian techniques have been developed in order to detect such structures in geophysical fluids. This is challenging because, while classical dynamical systems theory is defined for infinite time autonomous or periodic systems, in geophysical contexts the velocity fields are generally time-dependent, aperiodic in time, and defined over a finite discrete space-time domain. Among others, techniques developed are finite-size Lyapunov exponents (FSLE) (Aurell *et al.*, 1997), finite-time Lyapunov exponents (FTLE) (cf. Haller (2000); Haller and Yuan (2000); Haller (2001); Shadden *et al.* (2005)). Other techniques include distinguished hyperbolic trajectories (DHT) (Ide *et al.*, 2002; Ju *et al.*, 2003) and the direct calculation of manifolds as material surfaces (Mancho *et al.*, 2003, 2004, 2006b), the

geodesic theory of LCS (Haller and Beron-Vera, 2012) and the variational theory of LCS (Farazmand and Haller, 2012), etc.
Our choice in this work will be the use of the Lagrangian Descriptor (LD) function $M$ introduced by Madrid and Mancho (2009); Mendoza and Mancho (2010). The function $M$ has been applied in a variety of geophysical contexts. For example, in the ocean it has been used to analyze the structure of the Kuroshio current (Mendoza and Mancho, 2012), to discuss the performance of different oceanic datasets (Mendoza *et al.*, 2014), to analyze and develop search and rescue strategies at sea (Garcia-Garrido *et al.*, 2015), and to manage efficiently in real-time the environmental impact of marine oil spills (Garcia-Garrido *et al.*, 2016). In the field of atmospheric sciences, $M$ has been used to study transport processes across the Southern SPV and RWB by de la Cámara *et al.* (2012, 2013); Smith and McDonald (2014); Guha *et al.* (2016), and to investigate the Northern Hemisphere major stratospheric final warming in 2016 (Manney and Lawrence, 2016).

The dynamical system that governs the atmospheric flow is given by:

$$\dot{\mathbf{x}} = \mathbf{v}\left(\mathbf{x}(t),t\right) \ , \ \ \mathbf{x}(t_0) = \mathbf{x}_0 \ , \tag{5}$$

where $\mathbf{x}(t; \mathbf{x}_0)$ represents the trajectory of a parcel that at time $t_0$ is at position $\mathbf{x}_0$, and $\mathbf{v}$ is the wind velocity field. Since our interest is in the time scale of stratospheric sudden warmings ($\sim 10$ days) we can assume to a good approximation that the fluid parcels evolve adiabatically. Therefore trajectories are constrained to surfaces of constant specific potential temperature (isentropic surfaces). We will concentrate on the 850 K surface, which is in the middle stratosphere and approximately corresponds to the 10 hPa levels. In section 3 we expand on the reasons for this choice.

To compute fluid parcels trajectories it is necessary to integrate (5). As the velocity field is provided on a discrete spatio-temporal grid, the first issue to deal with is that of interpolation. We apply bicubic interpolation in space and third-order Lagrange polynomials in time (see Mancho *et al.* (2006a) for details). Moreover for the time evolution we have used an adaptive Cash-Karp method. It is important to remark that as done in (de la Cámara *et al.*, 2012) for the computation of particle trajectories we use cartesian coordinates in order to avoid the singularity problem arising at the poles from the description of the Earth's system in spherical coordinates. For our Lagrangian diagnostic we use the $M$ function defined as follows:

$$M(\mathbf{x}_0, t_0, \tau) = \int\limits_{t_0-\tau}^{t_0+\tau} \|\mathbf{v}(\mathbf{x}(t; \mathbf{x}_0), t)\| \, dt \ , \tag{6}$$

where $\| \cdot \|$ stands for the modulus of the velocity vector. At a given time $t_0$, the function $M(\mathbf{x}_0, t_0, \tau)$ measures the arc length traced by the trajectory starting at $\mathbf{x}_0 = \mathbf{x}(t_0)$ as it evolves forwards and backwards in time for a time interval $\tau$. Sharp changes of $M$ values (what we call singular features of $M$) occur for sufficiently large $\tau$, for very close initial conditions and highlight stable and unstable manifolds.

Mendoza and Mancho (2010, 2012) have performed systematic numerical computations of invariant manifolds and found that they are aligned with singular features of $M$. They also provide examples in geophysical flows where manifolds are defined in a constructive way. Invariant manifolds are mathematical objects classically defined for infinite time intervals. The unstable (stable) manifold of a hyperbolic fixed point or periodic trajectory is formed by the set of trajectories that in minus (plus) infinity time approach these special trajectories. In geophysical contexts this definition is not realizable, because only finite

time aperiodic data sets are possible. Nevertheless, manifolds can still be defined constructively with the following procedure. At the beginning time, these curves are approximated by segments with short length, aligned with the stable and unstable subspaces of the DHT identified with algorithms described in Ide *et al.* (2002); Madrid and Mancho (2009). This starting step aims to build a finite-time version of the asymptotic property of manifolds. Next segments are advected forwards and backwards in time by the velocity field. Due to the large expansion and contraction rates in the neighbourhood of the DHT, the curves grow rapidly in forwards and backwards time and specific issues are addressed by the procedure described in (Mancho *et al.*, 2003, 2004). The procedure provides curves, manifolds, that by construction are barriers to transport in geophysical flows. In this way since manifolds are aligned with singular features of $M$, the latter belong to invariant curves of the system (5), and therefore their crossing points are indeed trajectories of the system (5). The capability of LDs in general, and $M$ in particular, for revealing invariant manifolds was analyzed in detail in Mancho *et al.* (2013). Lopesino *et al.* (2015) and Lopesino *et al.* (2017) have discussed, in discrete and continuous time dynamical systems, respectively, a theoretical framework for some particular versions of LDs in specific examples.

The consistency between the output field of Eq. (6) and FTLE ridges has been discussed in some references (see Mendoza and Mancho (2010); de la Cámara *et al.* (2012); Mancho *et al.* (2013) ). The integral expression in Eq. (6) can be split in two terms: one for forwards time and other for backwards time integration. Explicit calculations discussed in Mancho *et al.* (2013) for a linear saddle, show that singular features of the first term are aligned with the stable manifolds while those for the backwards time integration are aligned with the unstable manifolds. This is similar to what is obtained with FTLE that highlight stable and unstable manifolds, respectively for forwards and backwards time integration intervals. The fact that we choose to add both fields is advantageous for highlighting hyperbolic trajectories at the crossing points of the singular features.

As an example relevant to the case that motivates the present study, we show in Fig. 1 the evaluation of $M$ over the Southern Hemisphere using $\tau = 15$ on the 850 K isentropic level for the 5th August 2002. The representation shows a stereographic projection (see Snyder (1987)) in which the SPV is clearly visible by the bright yellow color, and also the filamentation phenomena ejecting material both from the outer and inner part of the jet. These filaments are related to the presence of hyperbolic trajectories highlighted in the figure. The fact that these saddle points of the LD field are hyperbolic trajectories of the system (5) is numerically supported. To this end de la Cámara *et al.* (2013) show that (see their Fig. 2), for similar ERA-Interim fields, these points belong to the intersection of stable and unstable manifolds highlighted by the singular features of the field. In what follows, all figures showing $M$ were computed with $\tau = 15$. This choice of $\tau$ is made based on the fact that diabatic heating/cooling processes in the extratropical stratosphere generally have longer time scales than those of horizontal advection. Hence, air parcels move on two-dimensional isentropic surfaces to a good approximation (they stay within 850K for 30 days (Plumb, 2007). Moreover, diabatic heating rates in the Antarctic mid stratosphere are on the order of $0.5$ K day$^{-1}$, although uncertainties in this magnitude remain large (Fueglistaler *et al.*, 2009). During the time interval of our calculations of isentropic trajectories ($\tau = 15$ days, i.e. time period of 30 days), the material surface would experience an increase of potential temperature of around 15 K. Nevertheless, calculations of $M$ using wind fields at 850 K and 700 K (not shown) produce qualitatively similar results. This suggests that horizontal motions of the parcels will be affected by similar geometric structures at those isentropic levels and that the isentropic approach is justified in our problem.

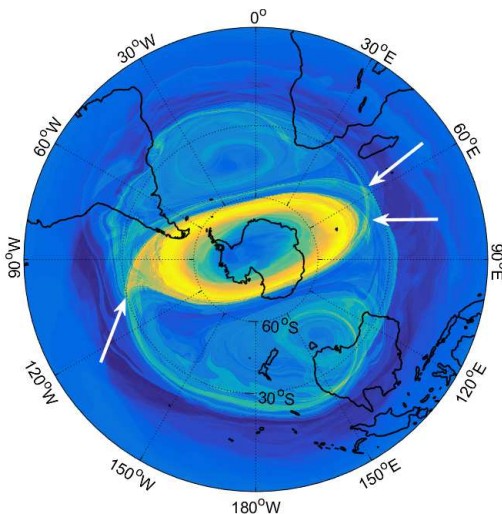

**Figure 1.** Stereographic projection of Lagrangian descriptors evaluated using $\tau = 15$ on the 850 K isentropic level for the 5th August 2002 at 00:00:00 UTC. The SPV is clearly visible as well as three hyperbolic trajectories (HTs) outside the vortex (marked with white arrows), two northeast and one southwest of it. Filamentation phenomena, which occurs in the neighborhood of HTs, is visible both inside and outside the vortex, where the outer filamentous structures play the role of eroding the jet barrier. Notice also the presence of two eddy coherent structures over the South Atlantic and south of Australia.

## 3   Data Analysis

As we indicated in the previous section, in order to characterize the planetary waves that propagate in the stratosphere we carry out a Fourier decomposition of the geopotential height. In Fig. 2 we show the axisymmetric mean-flow together with waves 1 and 2 in the geopotential field for the 22nd September 2002 on the 10 hPa pressure surface. The time evolution of these waves is described in the supplementary movies S1-S4. Animations S1-S3 show components 0, 1 and 2 separately for the time period of interest, while S4 shows the superposition of these three waves. It is important to reiterate that, since the geopotential provides a good approximation of the streamfunction of the large-scale flow in the extratropical regions, its analysis will provide us with guidance on the building of the simple kinematic model presented in the next section.

On the 10 hPa pressure level, the winter SPV in the Southern Hemisphere can be broadly defined as a circumpolar westerly jet. Figures 3a) and 3b) illustrate the evolution of the circulation during August-September 2002. We can clearly see the gradual deceleration of the SPV and the abrupt change in direction from westerly to easterly velocities at high latitudes that occurred on 22 September. This was a unique major SSW in the southern stratosphere. Planetary waves in the southern stratosphere were very active during the period where the 2002 SSW developed. Fig. 3 c) presents a time series of the ratio between the amplitudes of waves 1 and 2. Increased wave 1 amplitude results in a displacement of the SPV vortex from a circumpolar configuration, while increased wave 2 results in a stretching the SPV in one direction and contraction (or "pinching") in the

 orthogonal direction. According to Fig. 3 c), the amplitude of wave 1 was generally larger than that of wave 2 during the entire period, confirming the major role of this wave. Finally, Fig. 3 d) displays the variations in time of the ridges of wave 1 and wave 2. Note that wave 1 is quasi-stationary, while wave 2 propagates eastward as is typical in the southern stratosphere during early spring (Manney *et al.*, 1991; Quintanar and Mechoso, 1995).

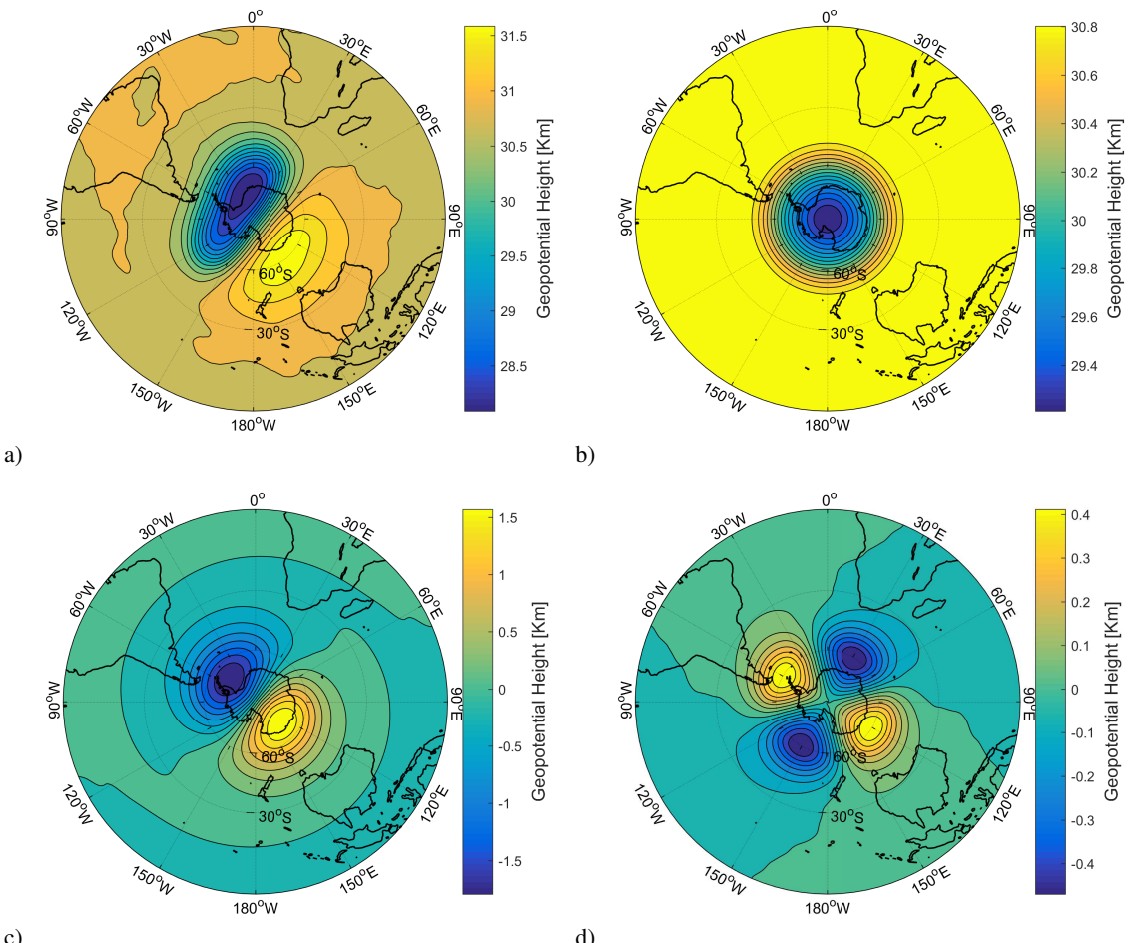

**Figure 2.** Stereographic projection of the geopotential height field and its Fourier decomposition for the 10 hPa pressure level on the 22nd September 2002 at 00:00:00 UTC: a) Geopotential height; b) Mean flow; c) Fourier component $\mathcal{Z}_1$; d) Fourier component $\mathcal{Z}_2$. Observe how the amplitude of the planetary wave with wavenumber 1 can be at least three times larger than that of wavenumber 2.

The contribution of these different waves to the evolution of the SPV and their transport implications is clearly observed in movie S5. A regime giving rise to the stretching of material lines and the appearance of hyperbolic regions and the associated filamentation processes is observed. These filamentous structures and HTs are clearly highlighted by the application of LDs to the wind fields, as shown in Figs. 1 and 4. Filamentation phenomena occurs both inside and outside the vortex, where the outer filamentous structures play the role of eroding the jet material barrier. Also, the presence of HTs in the flow (see captions of

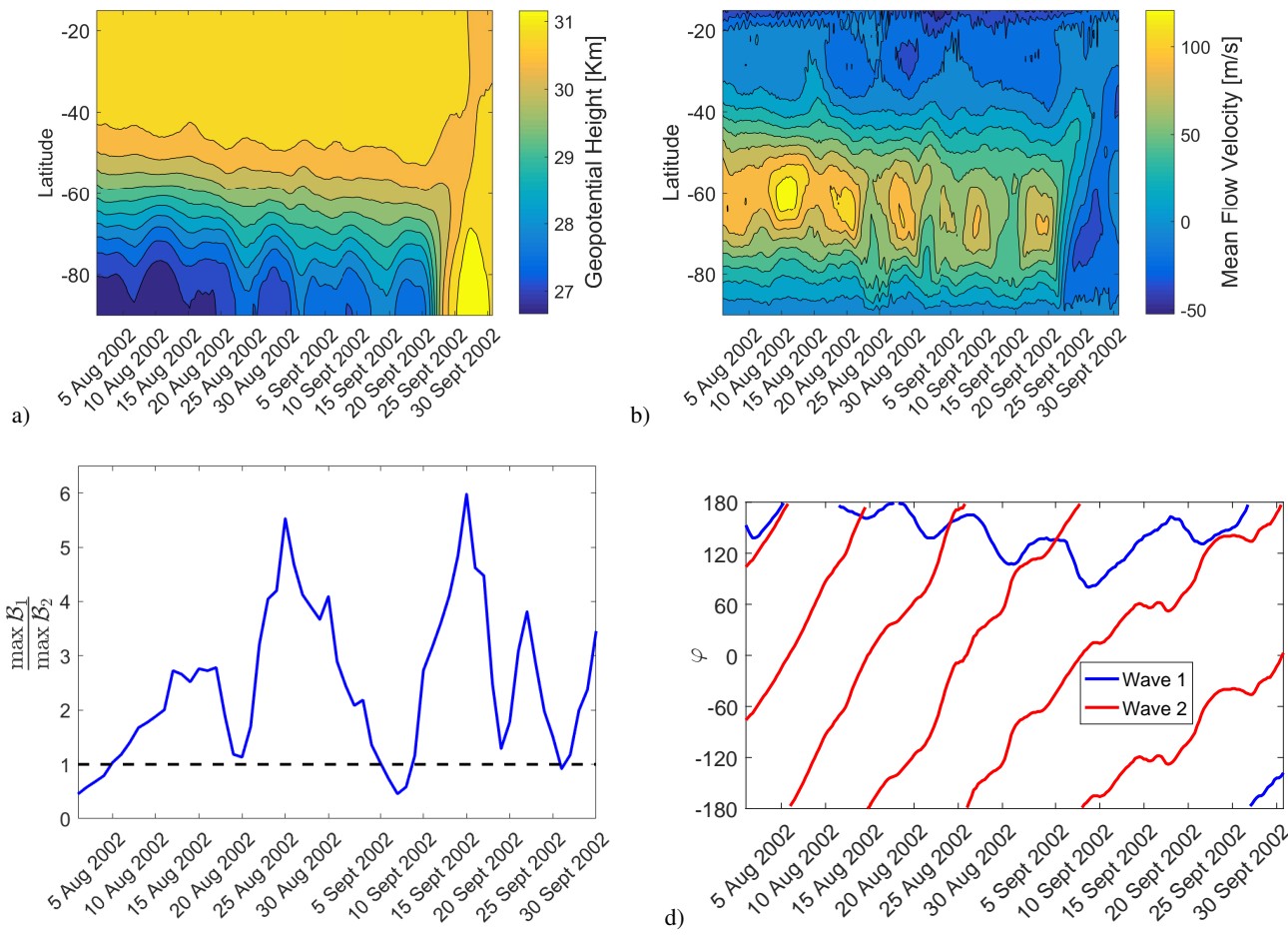

**Figure 3.** On the 10 hPa pressure level: a) Time evolution of the geopotential height corresponding to the mean flow. b) Time evolution of the mean flow velocity. Notice the change in wind direction from westerly to easterly that takes place from the 22nd to the 24th of September 2002. c) Time series of the ratio of the amplitudes of waves 1 and 2. d) Hovmöller (time-latitude) showing the position of the ridges of waves 1 and 2 at latitude $60°$S.

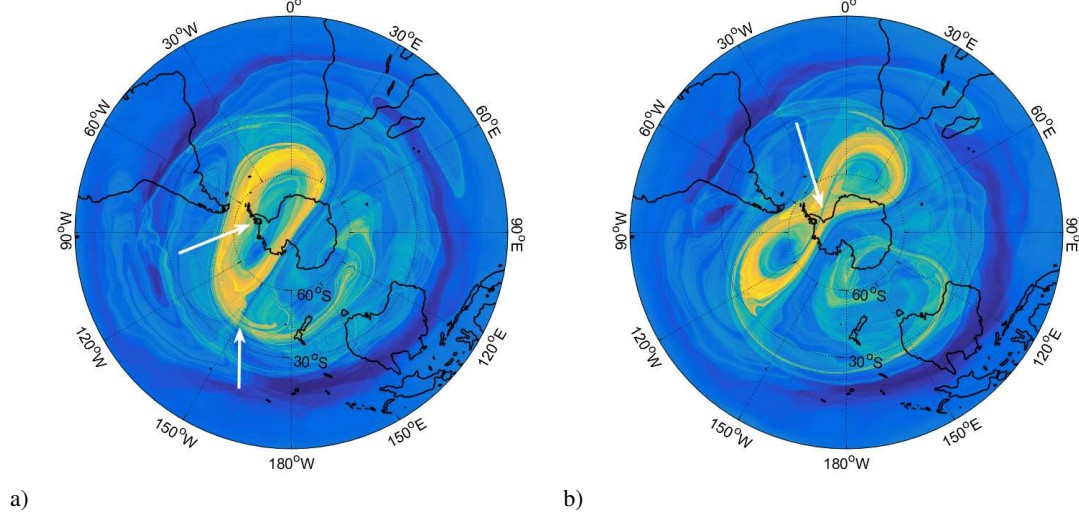

a)                                   b)

**Figure 4.** Stereographic projection of the $M$ function calculated using $\tau = 15$ on the 850 K isentropic level for the: a) 22nd September 2002 at 00:00:00 UTC; b) 24th September 2002 at 00:00:00 UTC. Filamentation phenomena and hyperbolic trajectories (marked with white arrows) are nicely captured in these simulations both in the exterior and the exterior of the SPV. Observe how the vortex breakdown on the 24th September occurs when, in the interior of the vortex, a HT allows the transport and mixing of parcels across the barrier

.

Figs. 1 and 4) indicate regions subjected to intense deformation and mixing (see Ottino (1989)). We emphasize that HTs appear
10   both inside and outside the SPV. Finally, the breakup of the SPV on the 24th September 2002 depicted in Fig. 4 b) (see also animation S5) occurs when manifolds associated with an HT that forms within the SPV connect the interior and the exterior of the jet, allowing for the interchange of parcels through the barrier. The pinching of the SPV takes place off the pole because $\mathcal{Z}_1$ has large amplitudes in the days preceding the breakup. As we approach the 24th September, $\mathcal{Z}_2$ becomes of the same order as $\mathcal{Z}_0$, and the jet elongates and flattens. At this point, the mean flow reversal is crucial for completing the pinching process and the appearance of a HT in the interior of the vortex as this splits apart.

5   **4   The kinematic model**

Kinematic models have a long history in the geophysical fluid dynamics community. They allow for a detailed parametric study of the influence of identified flow structures on transport and exchange of fluid parcels. All early studies utilizing the dynamical systems approach for understanding Lagrangian transport and exchange associated with flow structures such as meandering jets and travelling waves have employed kinematic models (see Samelson and Wiggins (2006)).
10   Continuing in this spirit, we propose a kinematic model that allows us to identify in a controlled fashion, the characteristics of the distinct propagating waves that are responsible for the different Lagrangian features observed in the SPV. Our kinematic model is inspired by the Fourier component decomposition of the geopotential extracted from the ERA Interim data as

discussed in the previous section. The analysis of data from August and September 2002 shows a mean axisymmetric flow, disturbed mainly by waves with planetary wavenumbers 1 and 2 whose amplitudes and phase speeds vary in a time-dependent fashion. Therefore we propose a kinematic model in the form of a streamfunction given by,

$$\Psi = \varepsilon_0 \Psi_0 + \varepsilon_1 \Psi_1 + \varepsilon_2 \Psi_2 , \tag{7}$$

where $\varepsilon_0, \varepsilon_1, \varepsilon_1$ are the perturbation parameters, which we will refer to as amplitudes, and $\Psi_i$ are the Fourier components along the azimuthal direction with wavenumbers $i = 0, 1, 2$ respectively, which we describe next.

We will work in a plane $(x, y)$ that is the orthographic projection of the Southern Hemisphere onto the equatorial plane (cf. Snyder (1987)). For simplicity, and in order to highlight the periodicity along the azimuthal direction, the components of the streamfunction are given in terms of polar coordinates satisfying $x = r\cos(\lambda)$ and $y = r\sin(\lambda)$ where the azimuthal direction $\lambda$ is related to the geographical longitude and $r$ is related to the geographical latitude.

The particular forms of $\Psi_0$, $\Psi_1$ and $\Psi_2$ are inspired by the Fourier decomposition of the geopotential field shown in figure 2 for the 10hPa pressure level on the 22nd September 2002. Starting with the mean zonal velocity, we will assume a jet with the following expression

$$v_\lambda = r(r-a)e^{-r}. \tag{8}$$

Therefore, $v_\lambda = 0$ only at $r = 0$ and $r = a$, and the velocity decreases exponentially away from the pole. Changing the values of $a$ will allow us to consider variations in the position of the jet maxima. Integration with respect to $r$ gives,

$$\Psi_0 = e^{-r}(ar + a - r(r+2) - 2), \tag{9}$$

The other streamfunction components are:

$$\Psi_1 = -r^2 e^{-r^2} \sin(\lambda) \tag{10}$$

and

$$\Psi_2 = (r/d)^2 e^{-r^2/d} \sin(2\lambda + \omega_2 t + \pi/4). \tag{11}$$

where $d$ and $w_2$ are also tunable constants, and the phase $\pi/4$ was added so that the relative positions of the waves 1 and 2 at $t = 0$ resemble those in figure 2. Positive values of $\omega_2$ correspondig to clockwide rotation. Note that (11) can represent a wave that propagates in the azimuthal direction $\lambda$ if $w_2$ is different than zero. Figure 5 shows the streamfunctions (9), (10) and (11) in the horizontal plane for the particular set of parameters indicated in the corresponding caption. In the panels of figure 5 and following, the center represent the South Pole and the circular dashed line indicates the Equator. The similary between

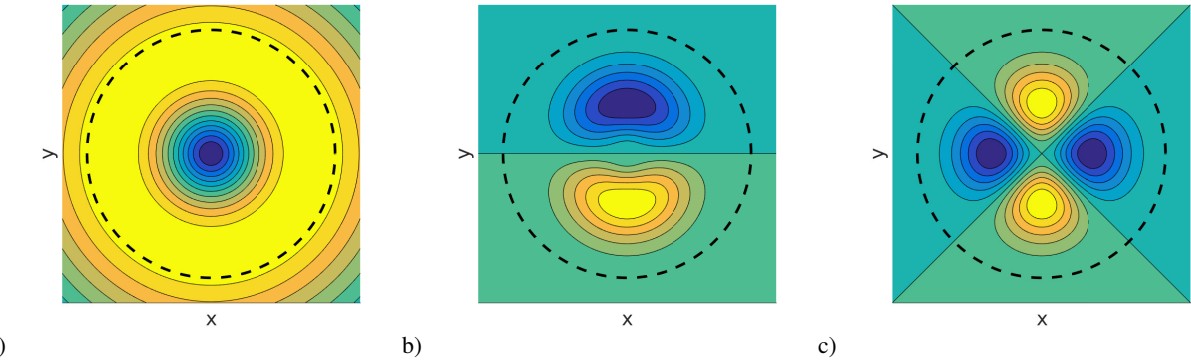

**Figure 5.** Representation of the three components of the streamfunction. a) $\varepsilon_0\,\Psi_0$, for $a=2$ and $\varepsilon_0 < 0$; b) $\Psi_1$; and c) $\Psi_2$ for $d=1$, $w_2 = 0$.

figure 2 and 5 for the selected set of parameters is evident taking into consideration that they correspond to stereographic and
orthographic projections, respectively.

The velocity of fluid parcels in the Cartesian coordinates $(x,y)$ is given by Hamilton's equations:

$$\frac{dx}{dt} = -\frac{\partial \Psi}{\partial y}, \;\; \frac{dy}{dt} = \frac{\partial \Psi}{\partial x} \tag{12}$$

We take the amplitudes to be time dependent in order to emulate changes in magnitudes. Let us start with $\varepsilon_0$ constant and,

$$\varepsilon_1 = \eta_1(1 + \sin(\mu t + \pi)), \;\; \varepsilon_2 = \eta_2(1 + \sin(\mu t)). \tag{13}$$

Here $\eta_1$ and $\eta_2$ are constants. The time dependence of $\varepsilon_1$ and $\varepsilon_2$ allows us to analyze each wave either separately or together
and their transient effect on the observed Lagrangian structures and therefore their transport implications. The time dependence
in (13) is such that one amplitude decreases when the other increases, roughly allowing conservation of the total energy when
both waves are present. In the simulations presented below $\mu = 2\pi/10$.

We begin by considering the case of a mean flow with $a=2$ and just wave 2 rotating at different speeds. Furthermore $d=1$
and $\eta_2 = 1$. Let us start with $\omega_2 = 0$, i.e. the stationary case. For this case, the dotted line in figure 6a) shows the azimuthal
velocity of the mean flow for $\varepsilon_0 = -2.5$, the dashed line is the azimuthal velocity of wave 2 at $\lambda = 0$, where the radial velocity
cancels, the solid line is the total azimuthal velocity and the blue line is the wave phase speed. According to figure 6a) there are
two points where the total velocity cancels, one being the origin. We can also easily see that there are additional fixed points
at the $r$ coordinate where the dotted and dashed curves intersect, but placed along the lines $\lambda = \pi/2, 3\pi/4$ . This gives at total
of five points in the hemisphere. Figure 6 b) shows the $M$ function for $\tau = 15$ evaluated on this stationary field at $t = 0$. The
minima of $M$ highlighting the five fixed points are evident. Moreover, we can see two two hyperbolic points in the outer part
of the vortex.

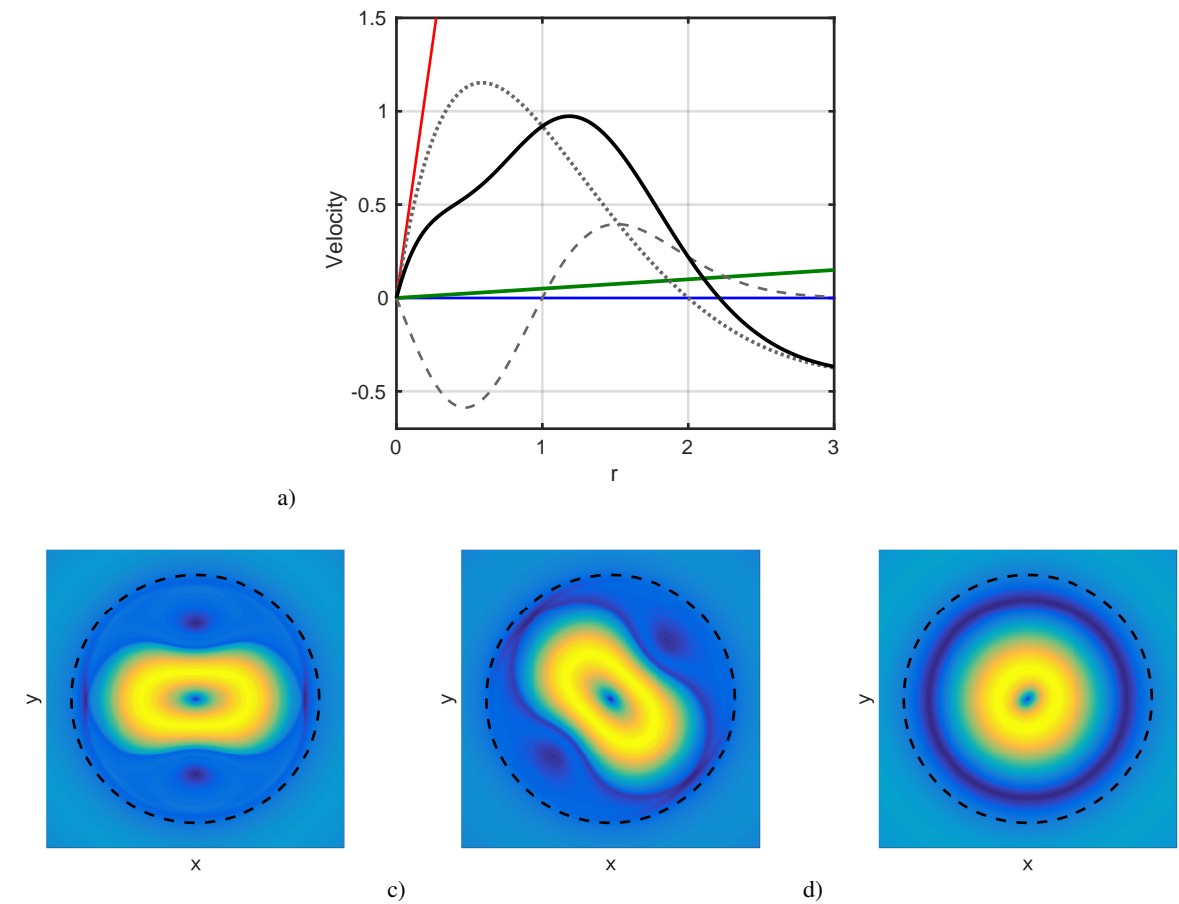

a)

b)                                      c)                                      d)

**Figure 6.** Some illustrative parameter choices for the kinematic model. a) A representation of the mean flow azimuthal velocity (dotted line), the azimuthal velocity of wave 2 for the stationary case along $\lambda = 0$ (dashed line), the total azimuthal velocity along $\lambda = 0$ (solid line), the phase velocity for $\omega_2 = 0.1$ (green line) and the phase velocity for $\omega_2 = 4\pi$ (red line); b) representation of the $M$ function for a kinematic model considering a mean flow ($a = 2$) plus a stationary wave 2 ($d = \eta_2 = 1$); c) the same as b) for a rotating wave 2 with $\omega_2 = 0.1$; d) the same as b) for a rotating wave 2 with $\omega_2 = 4\pi$.

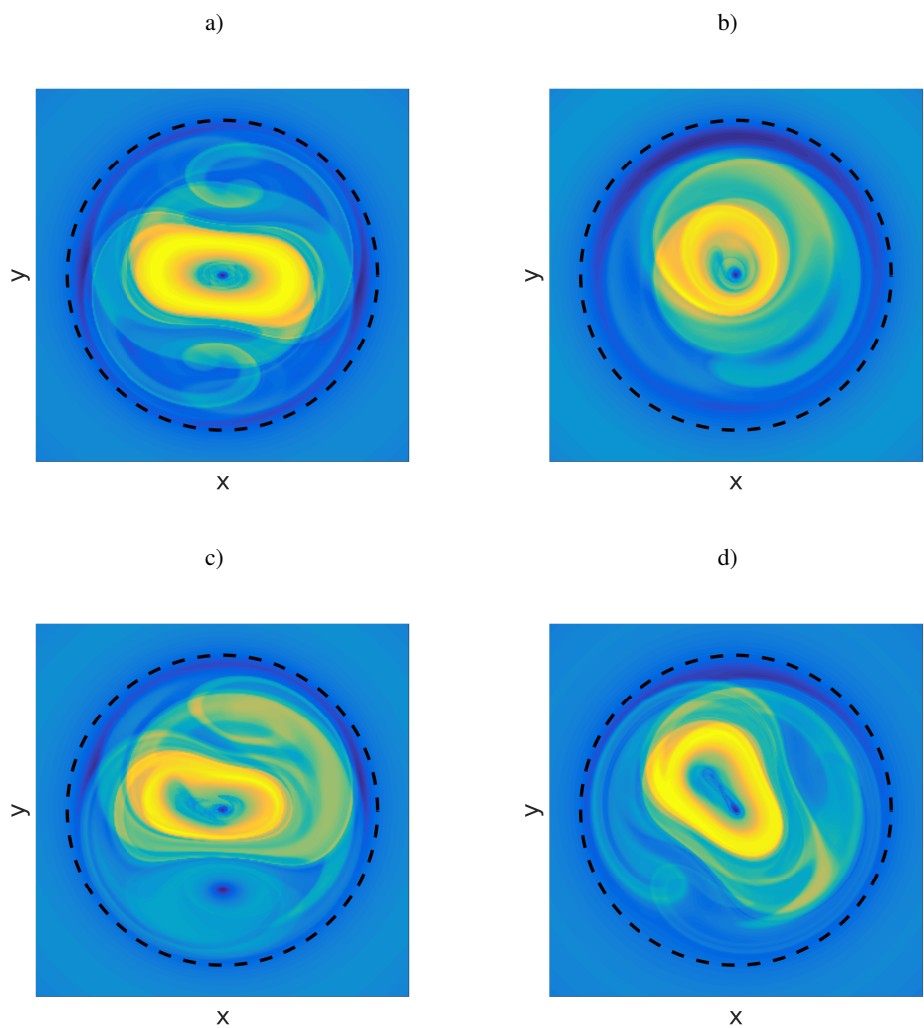

**Figure 7.** Lagrangian patterns obtained for $\tau = 15$ and different parameter settings in the kinematic model. a) Fourier components $\Psi_0$ and $\Psi_2$ the latter adjusted to perturb the vortex in its outer part; b) Fourier components $\Psi_0$ and $\Psi_1$ the latter adjusted to perturb the vortex in its outer part; c) the model keeps $\Psi_0$, $\Psi_1$ and $\Psi_2$. d) the model keeps $\Psi_0$, $\Psi_1$ and $\Psi_2$ with parameters adjusted differently to c);

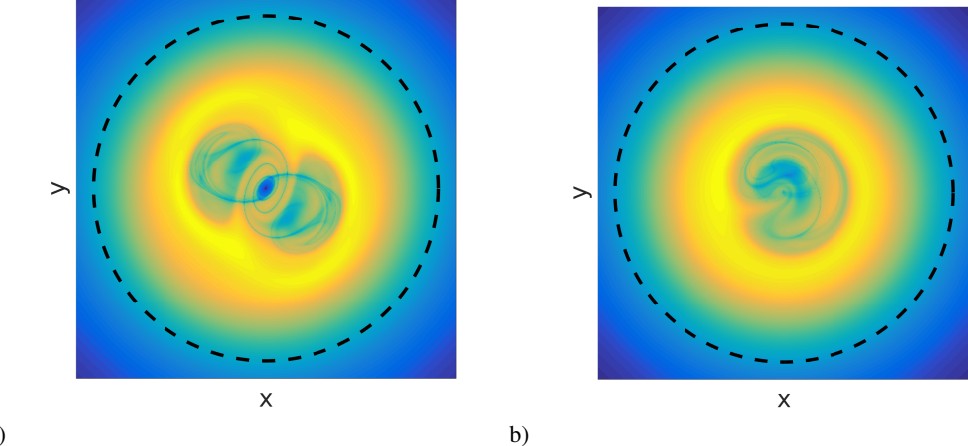

a)                                        b)

**Figure 8.** Lagrangian patterns obtained for $\tau = 15$ and different parameter settings in the kinematic model. a) the model keeps $\Psi_0$ and $\Psi_2$ adjusted to perturb the vortex in its interior part; b) The model keeps $\Psi_0$ and $\Psi_1$ adjusted to perturb the vortex in its interior part.

Next we consider the case with the same parameters except for $\omega_2$. Figure 6 c) shows how this picture changes when $\omega_2 = 0.1$, i.e. for slow rotation rate of wave 2. The total azimuthal velocity of the wave, in this case, is given by the dashed line in figure 6a) plus the phase velocity represented by the green line in the figure. If this total azimuthal velocity of the wave is added up to the mean flow two points are found in which the total azimuthal velocity cancels. Additionally, for a slow rotating wave, similarly to the previous case, the total azimuthal velocity of the wave can still be equal to the zonal mean velocity at some points in the domain. Therefore, figure 6c) is similar to figure 6b) except for a rotation. However, for a fast rotation of wave 2 ($\omega_2 = 4\pi$; red line), the total azimuthal velocity of the wave will be larger than the zonal mean velocity at all points in the domain. In this case, the pattern of $M$ (figure 6d) is very different from the pattern in figure 6b) showing no HTs.

Figure 7 displays the function $M$ obtained from the kinematic model for the same mean flow of figure 6a) and different parameters for the wave 1 and 2. All the representations are for $t = 0$ and $\tau = 15$. Figure 7 a) is for the same case as figure 6b), except that the amplitude of wave 2 changes in time ($\eta_1 = 0, \eta_2 = 1$). Again two HTs are visible in the external jet boundary along which filamentation occurs. Figure 7 b) corresponds to just wave number 1 changing amplitude in time ($\eta_1 = 1, \eta_2 = 0$). We can see one HT at the outer boundary of the jet where material of the vortex is being ejected. In these figures, transport processes producing filamentation ejecting material, have close connections to those present in Figures 1 and 4a), which have been linked to Rossby wave breaking at midlatitudes Guha *et al.* (2016). In figure 7 c) the mean flow is perturbed by the no rotating wave 2 of figure 7 a) and wave 1 of figure 7 b) ($\eta_1 = 1, \eta_2 = 1$). In figure 7 d) the parameters are the same as the figure 7 c), except that wave 2 rotates ($\omega_2 = 2\pi/15$). The jet shape and filamentary structures greatly resemble those present in the reanalysis data as shown in Figures 1 and 4a).

Figures 8 present a jet which in the interior is eroded by waves 2 and 1, respectively. To achieve such a configuration, free parameters are specifically tuned including a zonal mean flow with negative velocities near the pole. In figure 8 a) the mean

flow obtained with parameters $\varepsilon_0 = 2.6$ and $a = 0.75$ is perturbed by just a traveling wave 2 ($\eta_1 = 0, \eta_2 = 1, \omega_2 = -4\pi/25$)
with $d = 2$. Two filaments projecting material from the interior of the vortex are observed, and they are related to the presence
of interior HTs. In figure 8 b) the mean flow is obtained with the parameters $\varepsilon_0 = 2.5$ and $a = 0.5$. This mean flow is perturbed
by just a wave 1 with amplitude that varies in time ($\eta_1 = 1, \eta_2 = 0$). A protruding material filament from the interior of the
vortex is observed, which is related to the presence of an interior HT. The interior filaments in these figures recover features
that are identified as interior Rossby wave breaking phenomena in de la Cámara *et al.* (2013); Guha *et al.* (2016) and are also
visible from the reanalysis data as shown in Figures 1 and 4a).

Figure 4b) shows the pinching of the SPV in the observations on the 24th September 2002, which is before its breakup. In the
kinematic model, this structure can be obtained with a strong $\Psi_2$ and a substantial contribution from $\Psi_1$ to have a displacement
from the Pole. Movies S1, S2, S3 and S4 illustrate such structures. In order to reproduce the splitting we do not need to consider
the displacement and thus we neglect mode 1 in what follows. Figure 9 shows a sequence of $M$ patterns obtained with the
amplitude of mean flow is given by,

$$\varepsilon_0 = \eta_0(1 + \sin(\mu t + \pi)), \tag{14}$$

where $\eta_0 = -2.5$ and $\mu = 2\pi/10$, and a stationary wave 2 ($\omega_2 = 0$) with amplitude given by (13). Note that in this way the
mean flow weakens as wave 2 strengthens, and vice-versa. The parameters fit a streamfunction which a $t = 0$ coincides with
that used in figure 7a). The development of an hyperbolic point at the Pole in the observations (figure 4b)) can be clearly seen
in figure 9a). The two vortices have completely split at $t = 6$.

## 5 Kinematic models and conservation of Potential Vorticity

In this section we discuss the connection between the kinematic model introduced in the previous section and a fundamental
dynamical principle of geophysical fluids. Geophysical flows that are adiabatic and frictionless conserve the potential vorticity
$Q$ along trajectories. Conservation of $Q$ is expressed as follows:

$$\frac{dQ}{dt} = 0 \tag{15}$$

Here $d/dt$ stands for the material derivative. A natural question here is to discuss whether the proposed kinematic model
conserves $Q$. Let us assume that our setting is described by the quasigeostrophic motion of simple vortices in a shallow water
system (see Polvani and Plumb (1992); Nakamura and Plumb (1994)) in which $Q$ is given by:

$$Q = f_0 + \nabla^2 \Psi - \gamma^2 \Psi + f_0 \frac{h}{D} \tag{16}$$

Here $f_0$ is a constant related to the rotation rate, $D$ is the mean depth of the shallow water system, $D - h$ is the total depth, $h$ is
the bottom boundary of the fluid layer which is small when compared to $D$ and $\gamma = f_0/\sqrt{g_0 D}$, where $g_0$ is the gravity constant.
$\Psi$ is the geostrophic streamfunction for the horizontal velocity field, in our case given by expression (7), with parameters
corresponding to those of Figure 7 d), i.e. $\varepsilon_0 = -2.5$, $\eta_1 = 1$, $\eta_2 = 1$, $a = 2$, $d = 1$ and $\omega_2 = 2\pi/15$.

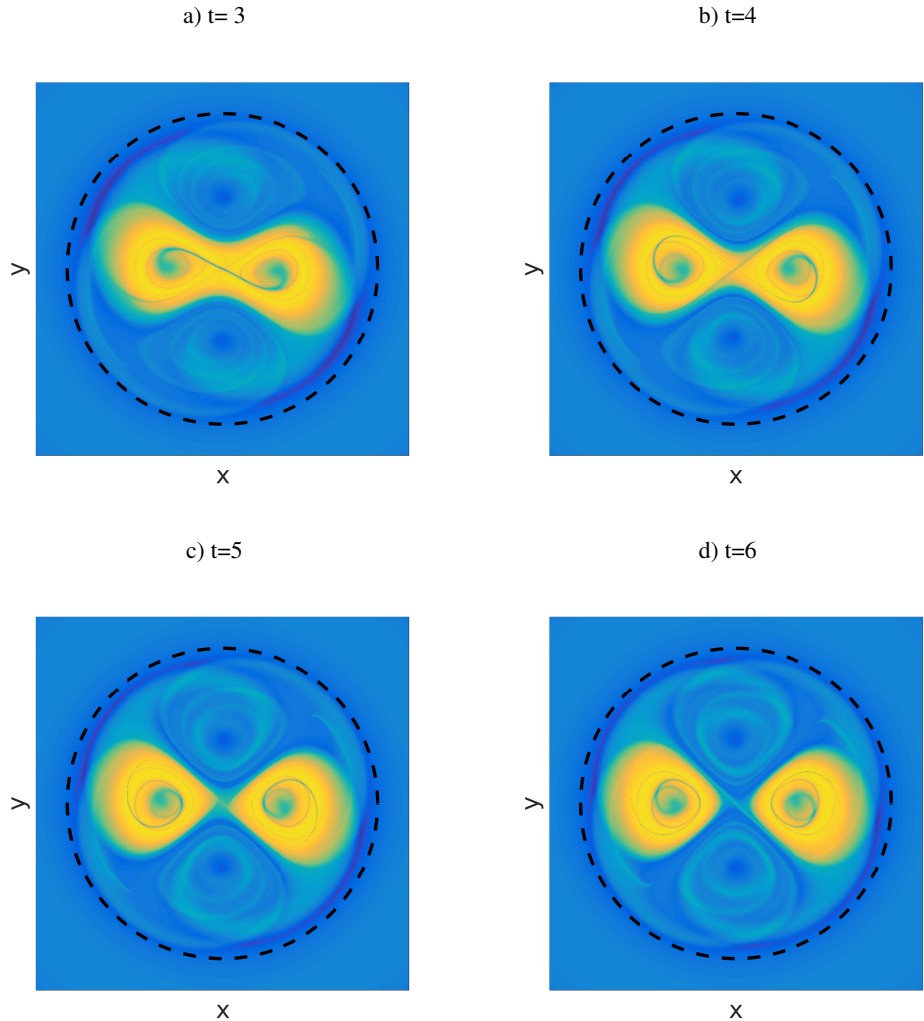

**Figure 9.** Evolution of the Lagrangian template for the case in which the mean flow decreases and the wave 2 increases. The sequence reproduces many of the Lagrangian features observed in the splitting event that occurred at the end of September 2002 (see movie S5). a) $t = 3$; b) $t = 4$; c) $t = 5$; d) $t = 6$.

We assume that at the initial time, $t = 0$, the vorticity $Q$ consist of a circular patch with constant vorticity $Q_0$ inside and vorticity $Q_1$ outside. At a later time $t = 2$, the vorticity distribution that preserves Eq. (15), is obtained by advecting the circular contour at $t = 0$ according to the motion equation (12), with algorithms described in Mancho *et al.* (2004). Figure 10 summarizes the evolution of the vorticity.

In order to preserve equation (16) from time $t = 0$, to time $t = 2$, and assuming the barotropic approach in which $\gamma = 0$, $h$ is solved from equation (16) as:

$$\frac{h}{D} = \frac{Q}{f_0} - \frac{\nabla^2 \Psi}{f_0} - 1 \tag{17}$$

Figure 11 shows the evolution of the function $h/D$ between $t = 0$ and $t = 2$. In particular the figure shows results for $Q_0 = 2$, $Q_1 = 1.8$ and $f_0 = 20$. We note that this calculation could have been repeated for any initial distribution of $Q$ defined as a piecewise constant function. The lower boundary $h$ is thus a time dependent function adjusted to preserve the conservation of the potential vorticity.. Without this forcing, kinematic models would not preserve potential vorticity.

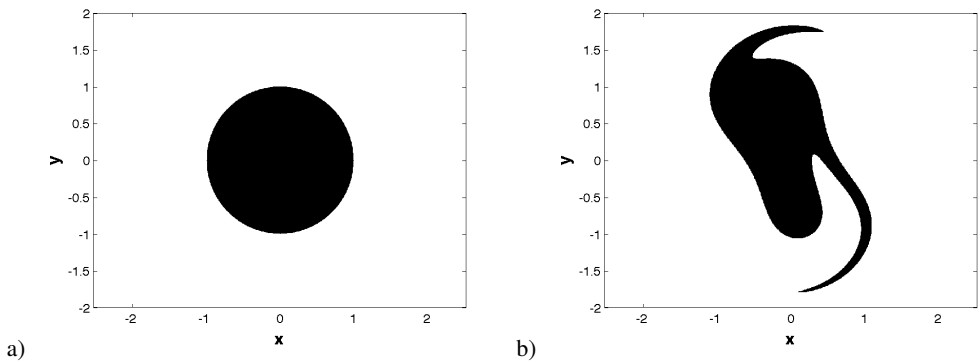

**Figure 10.** Evolution of a vorticity patch. a) Initial vorticity distribution at time $t = 0$; b) evolution of the vorticity at time $t = 2$.

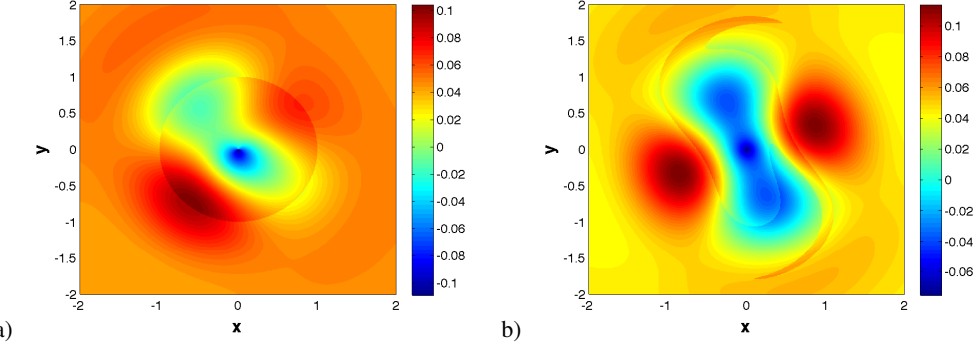

**Figure 11.** Evolution of the scaled lower boundary $h$. a) The function $h/D$ at time $t = 0$; b) evolution of $h/D$ at time $t = 2$.

## 6 Conclusions

In this work we propose a simple kinematic model for studying transport phenomena in the Antarctic Polar vortex. We are interested in gaining insights into the processes which carry material outwards from the vortex structure and inwards to the vortex structure.

The construction of the kinematic model is realized by analyzing geopotential height data produced by the ECMWF. In particular our focus is on the stratospheric sudden warming event that took place in 2002, producing the pinching and then breaking of the stratospheric polar vortex. We identify the prevalent Fourier components during this period, which consist of a mean axisymmetric flow and waves with wavenumbers one and two. The kinematic model is based on analytical expressions which recover the spatial structures of these representative Fourier components. The model can be controlled so that waves with wavenumbers one and two can be switched on and off independently. We are also able to adjust the relative position of the waves with respect to the mean axisymmetric flow.

The study of Lagrangian transport phenomena in the ERA-Interim reanalysis data by means of Lagrangian Descriptors highlights hyperbolic trajectories. These trajectories are Lagrangian objects 'seeding' the observed filamentation phenomena. The Lagrangian study of the kinematic model sheds light on the role played by waves in this regard. The model is adjusted to a stationary case which considers a mean flow and a stationary wave 2, that perturbs the mean flow in its outer part, producing hyperbolic trajectories. For the stationary case hyperbolic trajectories are easily identified. This framework is modified by transforming it to a time dependent problem by making the wave phase speed different from zero, or by introducing time dependent amplitudes. This allows to relate the time dependent structures with those easily identified in the stationary case. The setting is repeated with the wave 1, and both wave 1 and wave 2 together. The joint presence of these waves produces complex Lagrangian patterns remarkably similar to those observed from the analysis of the complex reanalysis data, and confirm the findings discussed by Guha *et al.* (2016). Further adjustement of some model parameters are able to produce erosion by means of filaments just in the interior part of the flow. Finally we point out that our analysis shows that the breaking and splitting of the polar vortex is justified in our model by the sudden growth of wave two and the decay of the axisymmetric flow.

*Acknowledgements.* V. J. García-Garrido, J Curbelo and A. M. Mancho are supported by MINECO grant MTM2014-56392-R. The research of C. R. Mechoso is supported by the U.S. NSF grant AGS-1245069. The research of S. Wiggins is supported by ONR grant No. N00014-01-1-0769. We also acknowledge support from ONR grant No. N00014-16-1-2492.

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
