# Peer review of "A Simple Kinematic Model for the Lagrangian Description of Relevant Nonlinear Processes in the Stratospheric Polar Vortex"

_Nonlinear Processes in Geophysics, 2016_

## Referee Comment (RC1) · Anonymous Referee #1 · 9 Jan 2017

In this paper, the authors construct a 2-D kinematic model of the Antarctic stratospheric polar vortex using the first three components of an axisymmetric stream function (i.e. a "mean flow", and waves with zonal wave number one and two). They highlight hyperbolic trajectories and their manifolds of such a flow by means of the so-called Lagrangian Descriptor or function M, and discuss the qualitative similarities to the Lagrangian structures obtained from atmospheric reanalyzed data (ERA-Interim) on isentropic surfaces during the major mid-winter warming of September 2002.

The paper is clear and well written, and most of the results are interesting, but the authors should do a much better job putting the paper and the results in context before meriting publication. I recommend publication after major revision.

Specific comments:

1. First paragraph (line 10-, page 1). The authors summarize the scientific findings that followed the discovery of the Antarctic ozone hole, but they do not cite any study at all (i.e. Chubachi 1984, Molina and Molina 1987, Bowman JAS 1993; JGR 1993, Manney et al. JAS 1994, etc.).

2. Page 2, line 16. "De la Camara et al (2013) suggested that HTs are representative of cat's eye structures...". McIntyre and Palmer (Nature 1983, JASTP 1984) and Bowman (JAS 1996) might be better references for this suggestion.

3. Page 2, line 25. "Our goal in the present study is to identify essential features in the filamentation process associated with the breakdown of the polar vortex..." I think the authors need to explain better the need for this study, putting it more in context. Why is this study interesting? Is this the first time anyone tries to show Lagrangian coherent structures during a sudden warming? What new insights into the dynamics of the polar vortex do you expect to gain from the analysis?

4. Page 4, line 15. Please cite some works as examples.

5. Page 5, line 26. The authors justify 2-D trajectories on the basis of isentropic motions with timescales of 10 days. If $\tau$ = 15 days, that means the trajectories expand $2\tau$ = 30 days. Is the 2-D motion approximation still valid? It would be useful to estimate the error growth of the 2-D trajectories (with respect to 3-D trajectories) with increasing $\tau$.

6. Figure 5, caption. "Notice the change in wind direction from westerly to easterly ... giving rise to the pinching of the SPV". The change in sign in zonal mean quantities does not reflect a particular change in the horizontal geometry of the vortex. Stratospheric warmings have been reported as displacement and split events (roughly wave-1 and wave-2 phenomena), but the zonal mean behavior of the zonal mean wind is rather similar. I would put it the other way round; it is the radical change in the vortex position and/or geometry during stratospheric warmings that gives rise to the change

in zonal mean wind direction.

7. Page 9, lines 3-5. "Finally, the breakup of the SPV on the 24th September 2002 depicted in Fig. 4 b) is caused by the formation of an HT in the interior of the vortex whose manifolds connect the interior and the exterior of the jet, allowing for the interchange of air through the barrier." From my point of view, the hyperbolic trajectory is a kinematic manifestation of a dynamical process. I am not sure if it is correct to state that the formation of the HT is the cause of the vortex breakdown.

8. Page 9, lines 7-8. $Z0$ is not independent of $Z1$ and $Z2$. In fact, linear theory states that the transient convergence of wave activity decelerates the mean flow, and this in turn affects the propagation and dissipation of the planetary waves.

9. Page 11, lines 16-18. In dynamically consistent models, those filaments could be related to wave breaking phenomena, or nonlinear vortex-vortex interactions. What is the reason for their presence in the kinematic model?

10. Page 13, lines 12-19 (Figure 7). I wonder if the amplitude reduction of $\Psi_0$ and amplification of $\Psi_2$ used to construct Fig. 7 is somewhat similar to what happened with $Z0$ and $Z2$ in the reanalysis data during the split event.

11. Section 5. It is possible that I have not followed the argument here. What are the values of $C$ and $h$ that you need to conserve $Q$ in your kinematic model? Are those values within the range of values used in shallow water models for the study of polar stratospheric dynamics?

Technical comments:

12. Figure 1, caption. "... coherent structures above and below the SPV". Please replace *above* and *below* with *over the South Atlantic* and *south of Australia*.

13. Figures 2 and 5 (and some movies). Please improve the color scale, the figures look blurry.

---

## Referee Comment (RC2) · Anonymous Referee #2 · 13 Jan 2017

**General comments:**

This paper addresses the issue of Lagrangian transport in the Stratospheric Polar Vortex (SPV). The first part of the paper analyzes SPV data from the ECMWF using the technique of Lagrangian Descriptors (LDs, developed over the years by some of the authors of this paper and their collaborators) for a specific time period in September 2002. A three-mode kinematic model which possesses the gross characteristics of the data is then developed, and there is some discussion on how it is possible by adjusting its parameters to mimic certain behaviors of the observational data. The paper is well-written and readable. However, I believe that some more work is needed to show that LDs are *relevant* to this situation, and that the kinematic model provides useful informa-

tion. I have expanded on this in my 'specific comments' below. My feeling is therefore that a major revision would be required before being acceptable for publication.

**Specific comments:**

1. It seems that the major focus is on modeling the SPV breakdown in September 2002. If trying to use Figure 4 as evidence that LDs provides an excellent way to explain this, then I feel that there must be some comparison to other studies which show this. Beyond a few brief references (page 2, line 27-28), the authors do not seem to do much in this direction. After all, how good are the results of Figure 4? What are the other symptoms of the SPV breakdown—what other observations showed that this indeed did break down? (Using Figure 3 is a start—but this is using an Eulerian observation to predict something Lagrangian—or is it?) And is Figure 4 consistent with any other observations? Several references which might help are: Nishii & Nakamura (*Geophys. Res. Lett.*, 2004), Kruger et al (*J. Atmos. Sci.*, 2005), Taguchi (*J. Atmos. Sci.*, 2014), Fisher et al (*Atmos. Chem. Phys.*, 2008), Esler & Scott (*J. Atmos. Sci.*, 2005), Konopka et al (*J. Atmos. Sci.*, 2005), Varotsos (*Environ. Sci. Pollution Res.*, 2002, 2003, 2004) and Allen et al (*Geophys. Res. Lett.*, 2003). In addition to these, I feel that it is imperative that there be comparisons (or relevant discussions) with the paper by Santitissadeekorn et al (*Phys. Rev. E*, 2010) which provides a Lagrangian analysis and provides pictures very similar to Figure 4.

2. The term 'Hyperbolic Trajectories' (HTs) is used often in this paper, and described briefly in the introduction. The ideas and intuition given in the third paragraph of the introduction are however only valid in *infinite-time* flows. There are sometimes additional limitations of steadines: the cat's-eyes structures in these models depends on drawing streamfunction contours (either in the steady frame or in a moving frame), and so are associated with steady situations. While the remainder of the discussion does not necessarily confine itself to steadiness, as far as I

am aware, hyperbolic trajectories can only unambiguously be defined for infinite-time situations, using the ideas of exponential dichotomies. The paper by Ide et al (*Nonlin. Proc. Geophys.*, 2002), for example, cites the exponential dichotomy definition—but this cannot be adequate for finite-time flows since the variational equation associated with *any* trajectory will obey the exponential decay requirement by choosing a suitably large prefactor. There have been attempts to fix this: by choosing a prefactor of $1$ (Doan et al (*J. Differential Equations*, 2012), Karrasch (*J. Differential Equations*, 2013), Duc & Seigmind (*Int. J. Bifurc. Chaos*, 2008)), or by extending to infinite-times in some fashion (Balasuriya, (*J. Nonlin. Sci.*, 2016)). In general, it seems that HTs are ill-defined for finite-time flows. Throughout the paper, however, the authors seem to be using 'saddle-like locations of the LD field' as their method of identifying HTs. I understand why such locations can be called 'hyperbolic,' but there does not seem to be any justification in calling them 'trajectories' since it is not at all clear if by following these in a time-varying way by computing LDs over a range of $t_0$ values, an actual *trajectory* of the system (5) arises. If the flow is nearly steady, it seems that it might be possible to establish the existence of time-varying saddle-points which are *close to* an actual (infinite-time) hyperbolic trajectory in some instances (Ide et al (*Nonlin. Proc. Geophys.*, 2002), Balasuriya, (*J. Nonlin. Sci.*, 2016)). But is this necessarily so for this situation, viz. using finite-time data, with moderate unsteadiness, and specifically using LD fields to identify saddle points? If the actual *term* 'hyperbolic trajectories' is not important to what the authors are doing, then perhaps they should simply call them saddle points of the LD field? But even so, claiming a direct relationship to stable and unstable manifolds—which are undefined for finite-time flows—seems problematic.

3. I have some concern about the centered nature of the definition for $M$ in (6). If requiring to find information on the 'skeleton of transport' at time $t_0$ using FTLEs/FSLEs/.../variational LCSs, the basic approach is to seed initial values at

$t_0$. If looking for the analog of the stable manifold at $t_0$ (i.e., repelling LCSs, ridges of forward-time FTLEs), these needs to be advected in forward time. Similarly, the advection is in backward time if looking for analogs of the unstable manifold. *It is this information which tells us about the skeleton at time $t_0$.* For example, Gaultier et al (*J. Marine Sci.*, 2013; *J. Geophys. Res. Oceans*, 2014) do this advection in backwards time in order to compare with sea-surface temperature fields at the time $t_0$. This is also because the advected scalar field (temperature in their case, whereas in this case it could be temperature, ozone concentration, etc, depending on the specific observable of interest in the SPV) at time $t_0$ would depend on the advection occurring *until the time $t_0$*. Future times surely cannot have an impact. Therefore, why is the integral in (6) being taken from times $t_0 - \tau$ to $t_0 + \tau$? This seems inconsistent with all other Lagrangian approaches. Moreover, it's hard to argue that the SPV knows the future! The pinch-off on September 24 in Figure 4(b), for example, uses velocity data into October.

4. The authors state that '$M$ reveal[s]/highlights Lagrangian coherent structures' (page 5, lines 12 and 15). Is there a rigorous justification for this—that $M$ specifically reveals *coherent* structures which move in a Lagrangian way according to the flow? If so, in what way? I am not able to find it directly in the cited references, though I am unable to get access to the latest article (Loposito et al, 2017) that is still in press. To my knowledge and judgment, a relationship has only been established in heuristic senses (and this is also so for other Lagrangian methods used and advocated by others), and in incredibly simplified test cases. Moreover, the authors talk of 'stable and unstable manifolds' here, but of course these things do not have a proper definition in finite-time flows. I believe that the description here needs to be watered down. The LD field is being used as a *heuristic*, and there is *some* evidence that it provides the right understanding.

5. The kinematic model requires more justification. Why do the amplitudes of the Fourier modes in the kinematic model have these particular $r$-dependencies?

The $r(r-a)$ in $v_r$ is understandable, but why $e^{-r}$? And why the specific forms chosen for $\Phi_1$ and $\Phi_2$? And why these particular forms of time-dependencies for $\epsilon_1$ and $\epsilon_2$? Certain parameter values are used in the simulations—why were these chosen? In what way are they consistent with parameter values of the SPV? Since the flow for the kinematic model is unsteady, the pictures of Figure 6 must be drawn at a particular time value $t_0$, I guess. What is it? I also have a much more general question regarding the kinematical model: What particular understanding does it give to this situation? It is probably possible to have the LD field display all sorts of crazy behavior by choosing the $\Phi$s in various ways, and so what does this particular model do? Now, if it was possible to argue, for example, that a particular instability arising from this kinematic model led to the SPV breakdown, then that might be interesting.

6. I am confused by what the authors are trying to achieve in Section 5. Are they trying to say (page 15, line 11) that their kinematic model can be made dynamically-consistent but inserting their $\Phi$ into (14) and (15) but then treating $h$ as unknown, and thereby getting an expression for $h$? This can possibly be done (though $h$ will satisfy a PDE which may not be easy to solve), but this is highly artificial. This would be demanding that the topography adjusts to the kinematic model that we insist is a solution. One possibility in which this part of the paper might have value is if the $\Phi$s in the kinematic model were somehow chosen as modes associated with the conservation equation (14)—this would be similar to the work of Pierrehumbert (*Geophys. Astrophys. Fluid Dyn.*,1991). The discussion of the earlier parts of this section also appears to lack relevance. If $Q$ were constant in patches, then complicated dynamics are possible subject to $Q$'s conservation—but this simply amounts to nullifying the dynamical constraint, and adds the extra condition (not talked about here) that the streamfunction needs to be chosen such that (15), for a constant $Q$, is satisfied. Basically, it is true that the potential vorticity distribution imposes constraints on the Lagrangian motion, which may

be an aspect the authors are trying to highlight here. For these, the papers by Brown & Samelson (*Phys. Fluids*, 1994) and Balasuriya (*Nonlin. Proc. Geophys.*, 2001), which deal with both constant and nonconstant $Q$, may be relevant. In general, I'm not sure I understand the goals this section, and so it requires some attention.

**Technical corrections:**

1. Some capitalization is missing in the references, for example in words like Rossby and Lagrangian.

2. Page 3, line 17: ERA needs to be capitalized, consistently with previous lines (e.g., line 12).

3. Page 3, line 20: space between 'fields' and 'on.'

4. Page 5, line 4: the citation to de la Cámara et al should not be within parentheses.

---

## Author Comment (AC1) · 7 Apr 2017

**Answer to Referee 1**

We wish to thank this referee for his/her very insightful comments. In our opinion, addressing these comments has helped us to strengthen the manuscript.

**Specific comments**

**1.** *1. First paragraph (line 10-, page 1). The authors summarize the scientific findings that followed the discovery of the Antarctic ozone hole, but they do not cite any study at all (i.e. Chubachi 1984, Molina and Molina 1987, Bowman JAS 1993; JGR 1993,*

*Manney et al. JAS 1994, etc.).*

We added "Solomon (1999; and references therein)". We also added "Chubachi, S., 1984a and Solomon" (1988; and references therein).

**2.** *Page 2, line 16. "De la Camara et al (2013) suggested that HTs are representative of cat?s eye structures . . . ". McIntyre and Palmer (Nature 1983, JASTP 1984) and Bowman (JAS 1996) might be better references for this suggestion.*

Thanks for pointing this out. We have added McIntyre and Palmer (1983) and Bowman (1996).

**3.** *Page 2, line 25. "Our goal in the present study is to identify essential features in the filamentation process associated with the breakdown of the polar vortex . . . " I think the authors need to explain better the need for this study, putting it more in context. Why is this study interesting? Is this the first time anyone tries to show Lagrangian coherent structures during a sudden warming? What new insights into the dynamics of the polar vortex do you expect to gain from the analysis?*

We have paid close attention to this comment. The Introduction starts with a new paragraph that addresses this concern. We state that the goal is to extract the physical mechanisms underlying notable transport features observed in complex data sets. We gain new insights into the fundamental mechanisms responsible for complex fluid parcel evolution, such as those associated with Rossby wave breaking phenomena, and describe a simple model having the ability to capture transport features, such as filamentation and vortex breaking. We have also added more discussion in the conclusions.

**4.** *Page 4, line 15. Please cite some works as examples.*

We have added Wiggins (2005) and Samelson and Wiggins (2006).

**5.** *Page 5, line 26. The authors justify 2-D trajectories on the basis of isentropic motions with timescales of 10 days. If $\tau = 15$ days, that means the trajectories expand $2\tau = 30$*

*days. Is the 2-D motion approximation still valid? It would be useful to estimate the error growth of the 2-D trajectories (with respect to 3-D trajectories) with increasing $\tau$. Page 5, line 26. The authors justify 2-D trajectories on the basis of isentropic motions with timescales of 10 days. If $\tau = 15$ days, that means the trajectories expand $2\tau = 30$ days. Is the 2-D motion approximation still valid? It would be useful to estimate the error growth of the 2-D trajectories (with respect to 3-D trajectories) with increasing $\tau$.*

We have added a paragraph addressing this comment at the end of section 2.2.

**6.** *Figure 3, caption. "Notice the change in wind direction from westerly to easterly ... giving rise to the pinching of the SPV". The change in sign in zonal mean quantities does not reflect a particular change in the horizontal geometry of the vortex. Stratospheric warmings have been reported as displacement and split events (roughly wave-1 and wave-2 phenomena), but the zonal mean behavior of the zonal mean wind is rather similar. I would put it the other way round; it is the radical change in the vortex position and/or geometry during stratospheric warmings that gives rise to the change in zonal mean wind direction.*

In view of these comments the description of the pinching was rewritten and extended at the end of section 4.

**7.** *Page 9, lines 3-5. "Finally, the breakup of the SPV on the 24th September 2002 depicted in Fig. 4 b) is caused by the formation of an HT in the interior of the vortex whose manifolds connect the interior and the exterior of the jet, allowing for the interchange of air through the barrier." From my point of view, the hyperbolic trajectory is a kinematic manifestation of a dynamical process. I am not sure if it is correct to state that the formation of the HT is the cause of the vortex breakdown.*

Throughout the manuscript we have replaced the expression "caused by the formation of a HT" by "occurred when a HT forms".

**8.** *Page 9, lines 7-8. Z0 is not independent of Z1 and Z2. In fact, linear theory states*

[Figure]

*that the transient convergence of wave activity decelerates the mean flow, and this in turn affects the propagation and dissipation of the planetary waves.*

The reviewer is correct. In the context of the kinematic model, the modes are given. We have based our specifications of the modes on the observation. The text have been revised to clarify this notion.

**9.** *Page 11, lines 16-18. In dynamically consistent models, those filaments could be related to wave breaking phenomena, or nonlinear vortex-vortex interactions. What is the reason for their presence in the kinematic model?*

The filaments mentioned by the referee, are related to the presence of hyperbolic trajectories that we link to wave breaking phenomena. In order to illustrate this in more detail we have rewritten section 4. Prior to the figure presenting the filaments mentioned by the referee, the kinematic model is adjusted to a stationary case, in which hyperbolic trajectories can be explicitly calculated as the velocity field is stationary (see new figure 6a)). Then the problem becomes non stationary by imposing a phase speed to the wave 2, and for slowly propagating waves hyperbolic trajectories are identified which are also rotating (see new figure 6b)). The pattern eventually produces filamentation in the pattern of $M$ (see figure 7 a)) by making in the kinematic model the amplitude of wave 2 time dependent.

**10.** *Page 13, lines 12-19 (Figure 7). I wonder if the amplitude reduction of $\Psi_0$ and amplification of $\Psi_2$ used to construct Fig. 7 is somewhat similar to what happened with Z0 and Z2 in the reanalysis data during the split event.*

Yes, we have selected perturbation amplitudes in accordance to the reanalysis data.

**11.** *Section 5. It is possible that I have not followed the argument here. What are the values of C and h that you need to conserve Q in your kinematic model? Are those values within the range of values used in shallow water models for the study of polar stratospheric dynamics?*

Section 5 has been rewritten and an explicit calculation of the forcing $h$ is reported, that achieves the conservation of potential vorticity $Q$ for one of the proposed $\Psi$. The calculation is illustrated for a simple $Q$ choice but it could be repeated for more realistic $Q$ distributions as far as they are defined as piecewise constant functions.

**Technical comments:**

**12.** *Figure 1, caption. "...coherent structures above and below the SPV". Please replace above and below with over the South Atlantic and south of Australia.*

Done

**13.** *Figures 2 and 5 (and some movies). Please improve the color scale, the figures look blurry.*

Contours were added to the figures. Thanks for pointing this out.

Please also note the supplement to this comment:
http://www.nonlin-processes-geophys-discuss.net/npg-2016-81/npg-2016-81-AC1-supplement.pdf

---

## Author Comment (AC2) · 7 Apr 2017

**Answer to Referee 2**

We wish to thank to this referee for his/her very useful comments that have helped us to improve the manuscript and have been addressed as follows:

**General comments:**

**1.** *This paper addresses the issue of Lagrangian transport in the Stratospheric Polar Vortex (SPV). The first part of the paper analyzes SPV data from the ECMWF using the technique of Lagrangian Descriptors (LDs, developed over the years by some of the*

*authors of this paper and their collaborators) for a specific time period in September 2002. A three-mode kinematic model which possesses the gross characteristics of the data is then developed, and there is some discussion on how it is possible by adjusting its parameters to mimic certain behaviors of the observational data. The paper is well-written and readable. However, I believe that some more work is needed to show that LDs are relevant to this situation, and that the kinematic model provides useful information. I have expanded on this in my specific comments below. My feeling is therefore that a major revision would be required before being acceptable for publication.*

We have clarified in a new version of the Introduction, the major goals of the article as maybe they were not sufficiently elaborated in original manuscript. The major goal is to gain new insights into the fundamental mechanisms responsible for complex fluid parcel evolution by providing a simple model (a kinematic model). The model allows in a controlled manner to recognize the physical mechanism responsible for the key observed transport features of SPV. In order to highlight the Lagrangian skeleton responsible for transport features both in the stratosphere and in the model, we use a Lagrangian tool, the function $M$, which has been extensively used in the literature. We consider that the references we provide in Section 2.2 provide a sufficient basis to use this tool, and we do not focus on justifying again in this new paper the efficiency of $M$ in highlighting Lagrangian features, we just use it.

**Specific comments:**

**1.** *It seems that the major focus is on modeling the SPV breakdown in September 2002. If trying to use Figure 4 as evidence that LDs provides an excellent way to explain this, then I feel that there must be some comparison to other studies which show this. Beyond a few brief references (page 2, line 27-28), the authors do not seem to do much in this direction. After all, how good are the results of Figure 4? What are the other symptoms of the SPV breakdown? what other observations showed that this indeed did break down? (Using Figure 3 is a start but this is using an Eulerian observation to predict something Lagrangian or is it?) And is Figure 4 consistent with*

*any other observations? Several references which might help are: Nishii & Nakamura (Geophys. Res. Lett., 2004), Kruger et al (J. Atmos. Sci., 2005), Taguchi (J. Atmos. Sci., 2014), Fisher et al (Atmos. Chem. Phys., 2008), Esler & Scott (J. Atmos. Sci., 2005), Konopka et al (J. Atmos. Sci., 2005), Varotsos (Environ. Sci. Pollution Res., 2002, 2003, 2004) and Allen et al (Geophys. Res. Lett., 2003). In addition to these, I feel that it is imperative that there be comparisons (or relevant discussions) with the paper by Santitissadeekorn et al (Phys. Rev. E, 2010) which provides a Lagrangian analysis and provides pictures very similar to Figure 4. ?*

The SPV breakdown in September 2002 has been extensively studied in the literature using ERA-Interim data and these references are now quoted in the manuscript. A novelty of our study is trying to understand the breakdown and its previous stages in a simple model that shows that the breakdown is related to wave propagation phenomena. The Lagrangian analysis of the breakdown exhibits what are the transport implications of the breaking, showing that the splitting leads to no mass transfer between the two vortices.

The paper by Santitissadeekorn et al (Phys. Rev. E, 2010) presents an interesting approach to estimating the three-dimensional location of the vortex. The promise of this approach is demonstrated by examination of the period from August 1 to September 31 in 1999. (The similarity of pictures during different final warming events can be expected from the similarity in evolution reported by Mechoso et al. (1988)). Our paper focuses on a different year (2002) and our concerns are not on the precise location of the polar vortex edge. Therefore, we will keep the paper the paper by Santitissadeekorn et al (Phys. Rev. E, 2010) in mind for future studies, but shall not include a reference in the text.

**2.** *The term "Hyperbolic Trajectories" (HTs) is used often in this paper, and described briefly in the introduction. The ideas and intuition given in the third paragraph of the introduction are however only valid in infinite-time flows. There are sometimes additional limitations of steadines: the cat's-eyes structures in these models de- pends on*

*drawing streamfunction contours (either in the steady frame or in a moving frame), and so are associated with steady situations. While the remain- der of the discussion does not necessarily confine itself to steadiness, as far as I am aware, hyperbolic trajectories can only unambiguously be defined for infinite- time situations, using the ideas of exponential dichotomies. The paper by Ide et al (Nonlin. Proc. Geophys., 2002), for example, cites the exponential dichotomy definition but this cannot be adequate for finite-time flows since the variational equation associated with any trajectory will obey the exponential decay requirement by choosing a suitably large prefactor. There have been attempts to fix this: by choosing a prefactor of 1 (Doan et al (J. Differential Equations, 2012), Karrasch (J. Differential Equations, 2013), Duc & Seigmind (Int. J. Bifurc. Chaos, 2008)), or by extending to infinite-times in some fashion (Balasuriya, (J. Nonlin. Sci., 2016)). In general, it seems that HTs are ill-defined for finite-time flows. Throughout the paper, however, the authors seem to be using "saddle-like locations of the LD field" as their method of identifying HTs. I understand why such locations can be called "hyperbolic", but there does not seem to be any justification in calling them "trajectories" since it is not at all clear if by following these in a time-varying way by computing LDs over a range of t0 values, an actual trajectory of the system (5) arises. If the flow is nearly steady, it seems that it might be possible to establish the existence of time-varying saddle-points which are close to an actual (infinite-time) hyperbolic trajectory in some instances (Ide et al (Nonlin. Proc. Geophys., 2002), Balasuriya, (J. Nonlin. Sci., 2016)). But is this necessarily so for this situation, viz. using finite-time data, with moderate unsteadiness, and specifically using LD fields to identify saddle points? If the actual term "hyperbolic trajectories" is not important to what the authors are doing, then perhaps they should simply call them saddle points of the LD field? But even so, claiming a direct relationship to stable and unstable manifolds "which are undefined for finite-time flows" seems problematic.*

We have extended the explanations on HTs in the Introduction and in Section 2.2. We provide references that compute and justify the use of HT in finite time data sets and also briefly summarize their content. In section 2.2 we have provided also references

and arguments that allow us to refer to the the "saddle-like locations of the LD field" as HTs. We have provided references in Section 2.2 that provide a constructive definition for finite time stable and unstable manifolds. We have also briefly summarized the content of these references in the text.

**3.** *I have some concern about the centered nature of the definition for M in (6). If requiring to find information on the ÔøΩskeleton of transportÔøΩ at time t0 using FTLEs/FSLEs/.../variational LCSs, the basic approach is to seed initial values at t0. If looking for the analog of the stable manifold at t0 (i.e., repelling LCSs, ridges of forward-time FTLEs), these needs to be advected in forward time. Similarly, the advection is in backward time if looking for analogs of the unstable manifold. It is this information which tells us about the skeleton at time t0. For example, Gaultier et al (J. Marine Sci., 2013; J. Geophys. Res. Oceans, 2014) do this advection in backwards time in order to compare with sea-surface temperature fields at the time t0. This is also because the advected scalar field (temperature in their case, whereas in this case it could be temperature, ozone concentration, etc, depending on the specific observable of interest in the SPV) at time t0 would depend on the advection occurring until the time t0. Future times surely cannot have an impact. Therefore, why is the integral in (6) being taken from times t0-tau to t0 + tau? This seems inconsistent with all other Lagrangian approaches. Moreover, itÔøΩs hard to argue that the SPV knows the future! The pinch-off on September 24 in Figure 4(b), for example, uses velocity data into October.*

In Section 2.2 we have included an explanation about the forward and backward integration time used for $M$, its relation with FTLE and the convenience of this choice for our study. Our approach is completely consistent with all other Lagrangian approaches, found in the literature.

**4.** *The authors state that "M reveal[s]/highlights Lagrangian coherent structures" (page 5, lines 12 and 15). Is there a rigorous justification for this - that M specifically reveals coherent structures which move in a Lagrangian way according to the flow? If so, in*

*what way? I am not able to find it directly in the cited references, though I am unable to get access to the latest article (Loposito et al, 2017) that is still in press. To my knowledge and judgment, a relationship has only been established in heuristic senses (and this is also so for other Lagrangian methods used and advocated by others), and in incredibly simplified test cases. Moreover, the authors talk of "stable and unstable manifolds" here, but of course these things do not have a proper definition in finite-time flows. I believe that the description here needs to be watered down. The LD field is being used as a heuristic, and there is some evidence that it provides the right understanding.*

There are rigorous justifications that invariant manifolds are aligned with singular features of LDs only for specific examples discussed in Lopesino et al 2015 for discrete dynamical systems and Lopesino et al. 2017 for continuous time dynamical systems. Also, the ability of LDs to highlight invariant sets has been explained, and the tool has been linked to the ergodic decomposition theory.

For geophysical flows Mendoza and Mancho (2010, 2012) have compared and found that numerically computed invariant manifolds systematically are aligned with singular features of $M$, but in these cases there is not any theorem supporting these facts, just numerical evidence. de la Cámara et al. (2013) show that for similar ERA-Interim fields, singular features of $M$ are aligned with numerically computed stable and unstable manifolds (see their Fig. 2).

These issues are explained now in Section 2.2

**5.** *The kinematic model requires more justification. Why do the amplitudes of the Fourier modes in the kinematic model have these particular r-dependencies? ? The r(r-a) in vr is understandable, but why exp(- r)? And why the specific forms chosen for PHI1 and PHI2? And why these particular forms of time-dependencies for eps1 and eps2? Certain parameter values are used in the simulations, why were these chosen? In what way are they consistent with parameter values of the SPV? Since the flow for*

*the kinematic model is unsteady, the pictures of Figure 6 must be drawn at a particular time value t0, I guess. What is it? I also have a much more general question regarding the kinematical model: What particular understanding does it give to this situation? It is probably possible to have the LD field display all sorts of crazy behavior by choosing the s in various ways, and so what does this particular model do? Now, if it was possible to argue, for example, that a particular instability arising from this kinematic model led to the SPV breakdown, then that might be interesting.*

Section 4 has been extensively revised to address the issues raised by the referee. In particular, the choice of free parameters in the kinematic model is explained in more detail. Further, the SPV breaking is reproduced by the kinematic model (see figure 9). The times at which specific patterns are achieved are also reported.

**6.** *I am confused by what the authors are trying to achieve in Section 5. Are they trying to say (page 15, line 11) that their kinematic model can be made dynamically-consistent but inserting their PHI into (14) and (15) but then treating h as unknown, and thereby getting an expression for h? This can possibly be done (though h will satisfy a PDE which may not be easy to solve), but this is highly artificial. This would be demanding that the topography adjusts to the kinematic model that we insist is a solution. One possibility in which this part of the paper might have value is if the s in the kinematic model were somehow chosen as modes associated with the conservation equation (14) this would be similar to the work of Pierrehumbert (Geophys. Astrophys. Fluid Dyn.,1991). The discussion of the earlier parts of this section also appears to lack relevance. If Q were constant in patches, then complicated dynamics are possible subject to Q's conservation? but this simply amounts to nullifying the dynamical constraint, and adds the extra condition (not talked about here) that the streamfunction needs to be chosen such that (15), for a constant Q, is satisfied. Basically, it is true that the potential vorticity distribution imposes constraints on the Lagrangian motion, which may be an aspect the authors are trying to highlight here. For these, the papers by Brown & Samelson (Phys. Fluids, 1994) and Balasuriya (Nonlin. Proc. Geophys.,*

*2001), which deal with both constant and nonconstant Q, may be relevant. In general, I'm not sure I understand the goals this section, and so it requires some attention.*

Section 5 has been rewritten and an explicit calculation of the forcing $h$ is reported, that achieves the conservation of potential vorticity $Q$ for one of the proposed $\Psi$. The calculation is illustrated for a simple $Q$ choice but it could be repeated for more realistic $Q$ distributions as far as they are defined as piecewise constant functions.

Please also note the supplement to this comment:
http://www.nonlin-processes-geophys-discuss.net/npg-2016-81/npg-2016-81-AC2-supplement.pdf
* * *